# Treatment of Critical-Size Femoral Bone Defects with Chitosan Scaffolds Produced by a Novel Process from Textile Engineering

**DOI:** 10.3390/biomedicines9081015

**Published:** 2021-08-14

**Authors:** Bruno Zwingenberger, Corina Vater, Roland L. Bell, Julia Bolte, Elisabeth Mehnert, Ronny Brünler, Dilbar Aibibu, Stefan Zwingenberger

**Affiliations:** 1University Center of Orthopedic, Trauma and Plastic Surgery, University Hospital Carl Gustav Carus, TU Dresden, 01307 Dresden, Germany; bzwingenberger@gmx.de (B.Z.); Corina.Vater@uniklinikum-dresden.de (C.V.); Julia.Bolte@uniklinikum-dresden.de (J.B.); Elisabeth.Mehnert@uniklinikum-dresden.de (E.M.); 2Center for Translational Bone, Joint and Soft Tissue Research, University Hospital Carl Gustav Carus, TU Dresden, 01307 Dresden, Germany; 3Department of Trauma-, Reconstructive- and Hand-Surgery, Municipal Hospital of the City of Dresden, 01067 Dresden, Germany; rolandlawrencebell@gmail.com; 4Institute of Textile Machinery and High Performance Material Technology (ITM), TU Dresden, 01062 Dresden, Germany; ronny.bruenler@tu-dresden.de (R.B.); dilbar.aibibu@tu-dresden.de (D.A.)

**Keywords:** chitosan, bone defect, tissue engineering, bone regeneration

## Abstract

The purpose of this study was to investigate, in vitro and in vivo, the suitability of chitosan (CHS) scaffolds produced by the net-shape-nonwoven (NSN) technology, for use as bone graft substitutes in a critical-size femoral bone defect in rats. For in vitro investigations, scaffolds made of CHS, mineralized collagen (MCM), or human cancellous bone allograft (CBA) were seeded with human telomerase-immortalized mesenchymal stromal cells (hTERT-MSC), incubated for 14 days, and thereafter evaluated for proliferation and osteogenic differentiation. In vivo, CHS, MCM and CBA scaffolds were implanted into 5 mm critical-size femoral bone defects in rats. After 12 weeks, the volume of newly formed bone was determined by microcomputed tomography (µCT), while the degree of defect healing, as well as vascularization and the number of osteoblasts and osteoclasts, was evaluated histologically. In vitro, CHS scaffolds showed significantly higher osteogenic properties, whereas treatment with CHS, in vivo, led to a lower grade of bone-healing compared to CBA and MCM. While chitosan offers a completely new field of scaffold production by fibers, these scaffolds will have to be improved in the future, regarding mechanical stability and osteoconductivity.

## 1. Introduction

The treatment of critical bone defects is a common problem that is encountered by physicians in operative orthopedics and trauma surgery in daily practice. The use of autologous bone graft represents the current gold standard for the mending of critical bone defects. However, obtaining the autologous graft is associated with much invasiveness, as well as prolongation of surgical time [1]. A synthetic, or semi-synthetic, substitute for this purpose, which can be readily and consistently produced, has been an objective in the field of tissue engineering [2].

Several studies show that the inexpensive, natural biopolymer chitosan (CHS), which can be harvested from the shell of crustaceans, exhibits a number of properties that promote bone growth [3,4,5]. In various in vitro, as well as in vivo, studies, the material was shown to be bioactive, biocompatible, non-toxic, biodegradable, non-immunogenic, as well as antibacterial. Furthermore, chitosan can be formed into a wide variety of three-dimensional shapes, for use as scaffolds for the treatment of critical bone defects, and is of relatively low cost. Studies have investigated chitosan-based scaffolds produced by freeze-drying, molding from a paste form, or even by particle aggregation [4,6].

The transformation of chitosan into a rigid structure has previously been described by Hild et al. [7], who developed the net-shape-nonwoven (NSN) technique via the wet spinning process, otherwise implemented in textile engineering. This method can produce scaffolds easily and quickly, with the pore size and porosity individually adapted to the defect, by the varying length and diameter of chitosan fibers [8]. Chitosan can also be processed to produce a thread that is suitable for weaving or knitting, to manufacture sewn or knitted structures with a higher biomechanical stability and elasticity. These structures might be used to adapt bony structures to soft tissues, such as tendons or muscles.

Previously established scaffolds, made of mineralized collagen (MCM) and cancellous bone allograft (CBA), provide suitable comparison groups.

The aim of this study was to investigate the influence of scaffolds that were made of CHS, by the NSN method, on the healing of critical-size bone defects. In vitro data were generated to investigate the scaffolds’ effect on human telomerase-immortalized mesenchymal stromal cells (hTERT-MSC), with respect to proliferation, maintenance of cell viability, and osteogenic differentiation. Bone healing, in vivo, was evaluated radiographically and histologically 12 weeks after implanting CHS, MCM or CBA scaffolds into 5 mm femoral bone defects in 10-week-old Wistar rats.

## 2. Materials and Methods

### 2.1. Study Design

In vitro, the scaffolds were seeded with hTERT-MSC and examined for cell viability and proliferation, as well as osteogenic differentiation after an incubation period of 14 days.

In vivo, 32 male 10-week-old rats were randomly distributed to 3 experimental groups (MCM, CBA, CHS). Then, 5 mm critical size bone defects were created in the right femur of the animals and stabilized by an internal fixator. Thereafter, CHS, MCM and CBA scaffolds were implanted. After an observation period of 12 weeks, the rats were sacrificed and the femora were explanted for µCT and histological analysis.

### 2.2. Scaffolds

#### 2.2.1. CHS Scaffolds Made by NSN Technique

The chitosan scaffolds were prepared using the NSN technique described by Hild et al. [1]. For this method, short fibers are bonded at their intersections—layer by layer—by use of a solvent. The technique allows for production of scaffolds with wide variability in both geometry and porosity. The high overall porosity allows for a good ratio of volume-to-functional surface area, which favors cell adhesion, migration and proliferation. An advantageous aspect of using the NSN technique is that by varying the length and diameter of the fibers, the porosity and pore size of the scaffold can be adapted to the required biological conditions, e.g., to optimize the osteoconductive properties [7]. Figure 1A shows the microstructure of the CHS scaffolds.

The scaffolds were fabricated at the Institute of Textile Machinery and High Performance Textile Materials Engineering (TU Dresden, Dresden, Germany). Chitosan consisting of 90% deacetylated chitin (Heppe Medical Chitosan GmbH, Halle, Germany) with a molar mass of 200 to 300 kDa was used for the fabrication process. The chitosan was dissolved in diluted acetic acid, and wet spinning was then used to produce fibers with a diameter of 40 µm and a length of 1 mm. These fibers were then joined by use of acetic acid to form three-dimensional scaffolds of globular shape with 4 mm diameter for in vitro studies and of cylindrical shape with 8 mm length and 6 mm diameter for in vivo studies. The scaffolds had a calculated porosity of 89.41 ± 0.18% (voxel-based modeling software GeoDict; Math2Market GmbH, Kaiserslautern, Germany) and a pore size of 104.83 ± 6.83 µm (PSM 165, Topas GmbH, Dresden, Germany) [8]. Before use in the experiments CHS scaffolds were gamma-sterilized with 25 kGy.

For in vivo investigations, the scaffolds had to be stabilized by coating them with a fibrin gel (Tisseel, Baxter International Inc., Deerfield, IL, USA), which improved mechanical stability sufficiently for the implantation.

#### 2.2.2. CBA Scaffolds

Scaffolds made of human cancellous bone (CBA, DIZG, Berlin, Germany) were filed into a cylindrical shape with a diameter of 5.8 mm and a length of 4.5 mm. The side facing the internal fixator was filed flat to ensure an optimal fit. The scaffolds were then cleaned in an ultrasonic bath. Before implantation, CBA scaffolds were sterilized by steam autoclaving in water at 121 °C. The scaffolds were then dried overnight under a laminar-flow hood at room temperature. The microstructure of CBA scaffolds can be observed in Figure 1C.

#### 2.2.3. MCM Scaffolds

Cylindrical scaffolds made of fibrillated mineralized collagen (Ø 4 mm, length 8 mm), with a mean pore-size of around 200 µm were produced as described previously [9]. Before use in the experiments MCM scaffolds were gamma-sterilized with 25 kGy. An overview of the microstructure of MCM scaffolds is depicted in Figure 1B.

### 2.3. In Vitro Investigations

#### 2.3.1. Cells

A human mesenchymal stem cell line expressing human telomerase reverse transcriptase (hTERT-MSC) [10], kindly provided by Matthias Schieker (Laboratory of Experimental Surgery and Regenerative Medicine, University Hospital Munich (LMU, München, Germany)), was used for the in vitro experiments.

#### 2.3.2. Cell Seeding & Cultivation

Before seeding with cells, CBA and MCM scaffolds were equilibrated with basic medium (Dulbecco’s modified Eagle medium, 10% fetal calf serum, 1% penicillin/streptomycin (Sigma-Aldrich, St. Louis, MO, USA)) for 24 h.

Filter paper was used to absorb excessive medium from CBA and MCM scaffolds; CHS scaffolds were carefully moistened with basic medium. After transferring the scaffolds in wells that were lined with a hydrophobic foil (PARAFILM, Sigma-Aldrich), 1 × 10^5^ hTERT-MSC were seeded in triplicates on top and allowed to adhere at 37 °C and 5% CO_2_. After 2–3 h, scaffolds were carefully transferred into new wells, filled up with basic medium and incubated for up to 14 days. Cells seeded at 1 × 10^4^/cm^2^ in triplicates on tissue culture polystyrene served as control.

On day 3, osteogenic differentiation of the cells was induced by supplementing the medium with 100 nM dexamethasone, 10 mM beta-glycerophosphate, 50 μM ascorbic acid-2-phos-phate and 10 nM vitamin D_3_ (all from Sigma-Aldrich).

#### 2.3.3. Analysis of Cell Proliferation and Osteogenic Differentiation

After 14 days of incubation, samples were washed with phosphate-buffered saline (PBS) and then frozen at −80 °C until analysis was conducted.

To investigate cell proliferation and osteogenic differentiation, determination of DNA content (proliferation) and alkaline phosphatase (ALP, osteogenic differentiation) activity was carried out. For this purpose, frozen samples were thawed and lysed with 1% Triton X-100/PBS (Merck, Darmstadt, Germany) for 50 min on ice. Cell numbers were then determined using the QuantiFluor^®^ dsDNA system (Promega, Madison, WI, USA) according to the manufacturer’s instructions. Fluorescence was quantified using a microplate reader (infinite M200 PRO, Tecan, Switzerland) and correlated to a calibration line generated from defined numbers of cells from the same experiment.

To measure ALP activity, cell lysates were incubated with 1 mg/mL p-nitrophenylphosphate in 0.1 M diethanolamine (pH 9.8) containing 1% Triton X-100 and 1 mM MgCl_2_ at 37 °C for 30 min (all from Merck). After stopping the enzymatic reaction by adding 1 M NaOH (Merck), absorbance was measured at 405 nm. Values were correlated to a calibration line generated from different dilutions of a 1 mM p-nitrophenol stock solution (Merck) and normalized to the cell number.

### 2.4. In Vivo Investigations

#### 2.4.1. Animals

For the in vivo study 32 male 10-week-old Wistar rats (weight: 404.8 ± 15.1 g, range 370–428 g) were purchased from Janvier Labs (Le Genest-Saint-Isle, France) and randomized into the following 3 groups: (1) CHS (*n* = 11), (2) CBA (*n* = 11) and (3) MCM (*n* = 10).

While kept at 2 animals/cage in a 12 h light—dark cycle, animals were fed a standard diet with food and water ad libitum. All animal experiments were performed in accordance to the National Institutes of Health Guidelines for the Use of Experimental Animals and were approved by the Local Animal Care and Ethics Committee of Dresden University Hospital (protocol no. DD24-5131/354/10; approved 29 April 2016).

#### 2.4.2. Surgical Procedures

The animals were anesthetized using 2.5–3% isoflurane (1 L/min in 100% O_2_). For analgesia, each animal received 2 mL saline containing 12 µg buprenorphine subcutaneously.

Surgeries were performed in prone position as described previously [11,12]. The right hind limb of the animal was shaved and disinfected, and an approximately 3 cm skin and fascia incision was made along the length of the right femur. After freeing the right lateral femur from soft tissue, a customized 5-hole internal fixation plate (Ø 1.5 mm straight locking plate, stainless steel; LCP Compact Hand System, DePuy Synthes, West Chester, PA, USA) was fixed to the femur using 2 proximal and 2 distal screws (Ø 1.5 mm locking screws, stainless steel; length outer screws: 7 mm, length inner screws: 6 mm; DePuy Synthes). A custom-made 3-dimensional printed saw guide was then placed on the lateral femur above the third empty hole of the plate and a 5 mm osteotomy was performed using 2 Gigli wires (0.44 mm; RISystem AG, Landquart, Switzerland). The resected mid-diaphyseal bone was carefully removed and the wound was thoroughly rinsed with saline to remove metallic remnants of the Gigli wires. The 5 mm defect was then filled with a CHS, CBA or MCM scaffold (Figure 2).

Soft tissue was then returned to its anatomical position and the wound was closed in a layered fashion (muscles: single button resorbable suture, Vicryl 4-0, Ethicon, Raritan, NJ, USA; skin: Donati non-resorbable suture, Ethilon 4-0, Ethicon). After wound closure, animals were transferred back into their cage and allowed to wake up under an infrared lamp.

#### 2.4.3. Preparations for Micro-Computed Tomography (µCT) and Histology

After 12 weeks, anesthetized rats were euthanized by CO_2_ asphyxiation and both femora (defect and contralateral) were harvested while ensuring that no damage to new callus resulted.

The internal fixators were carefully removed, and the femora placed in 4% neutral buffered formaldehyde (SAV Liquid Productions, Flintsbach am Inn, Germany), which was changed every 48 h.

#### 2.4.4. High-Resolution µCT

For µCT measurement, fixed femora from each specimen were wrapped with cling film, individually placed in a test tube, and scanned with a µCT (vivaCT 40, Scanco Medical, Wangen-Brüttisellen, Switzerland) using the following settings: X-ray intensity = 114 µA, X-ray tube = 70 kVp, voxel size = 21 µm, integration time = 200 ms.

The most proximal and distal screw holes were used to define the micro-CT scanning area, which corresponded to a volume of interest of approximately 11 mm (525 slices). Micro-CT image sets from each sample were analyzed using the SCANCO Medical software suite. An approximately 8 mm (380 slices, starting from the proximal and distal drill-holes nearest the defect) cylindrical region of interest was defined to quantify the bone regenerated in the defect (BV in mm^3^). To identify bone, a threshold of 200 mg hydroxyapatite/cm^3^ was used for all specimens. 

#### 2.4.5. Histology

After scanning by µCT, the femora were decalcified for 14 days in ethylenediaminetetraacetic acid (EDTA), which was changed every 48 h. Bones were dehydrated by an ascending ethanol series, embedded in paraffin, and then cut into 2 µm sections. Deparaffinization and hydration were performed using standard procedures before staining the slides. All histological sections were evaluated using a BIOREVO BZ-9000 microscope (KEYENCE Deutschland GmbH, Neu-Isenburg, Germany).

To evaluate the grade of defect healing, the sections were stained with hematoxylin and eosin (H&E; Merck, Darmstadt, Germany). Three representative sections per animal were analyzed by 3 different, blinded observers. Subsequently, the sections were classified according to Huo et al. [13] (Table 1).

To obtain a better overview, sections were additionally stained with Goldner’s trichrome (Merck, Darmstadt, Germany).

Sections examined for vascularization were stained for α-smooth muscle actin (rabbit anti-smooth muscle actin, clone 1A1, 1:750, Agilent Dako, Santa Clara, CA, USA). Each blood vessel with visible lumen was counted. 

Osteoblasts were quantified by immunohistochemical staining for bone alkaline phosphatase (BAP; rabbit anti-BAP, 1:100, LINARIS Biologische Produkte GmbH, Dossenheim, Germany) and appeared as brown-stained cuboidal mononuclear cells when using 3,3′-Diaminobenzidin (DAB) as chromogen [14].

Presence of osteoclasts was analyzed using immunohistochemical staining for tartrate-resistant acid phosphatase (TSP; naphthol-ASBI substrate, Sigma-Aldrich). TSP is typically found on osteoclasts and has been established as a suitable marker for these cells, which present morphologically with a ruffled boarder and are located in lacunae [15,16].

### 2.5. Statistics

All data are presented as mean ± standard deviation. Statistical analysis was performed using GraphPad Prism 5 (GraphPad Software, San Diego, CA, USA). Statistical significance was designated for *p* ≤ 0.05. Analysis of overall significance was performed using one-way ANOVA following Tukey’s post hoc testing to compare normally distributed individual groups.

## 3. Results

### 3.1. In Vitro Cell Proliferation and Osteogenic Differentiation

The quantitative measurement of DNA revealed significant differences between all the groups (one-way ANOVA, *p* = 0.014). Compared to CHS, there was a significantly higher number of cells observed on the scaffolds of the CBA group (6.25 ± 0.73 × 10^4^ cells/well, *p* < 0.05) and the MCM group (5.29 ± 0.81 × 10^4^ cells/well, *p* < 0.05) compared with CHS (2.45 ± 1.59 × 10^4^ cells/well, *p* < 0.05), after comparison by the test according to Tukey, as shown in Figure 3A.

After 14 days, the amount of alkaline phosphatase per 10^4^ cells was measured. Regarding the overall significance, determined by one-way ANOVA, strong differences were shown (*p* = 0.001). The CHS group (0.019 ± 0.004 µmol/10^4^ cells) showed significantly better osteogenic differentiation compared to both the MCM (0.005 ± 0.002 µmol/10^4^ cells, *p* < 0.01) and CBA groups (0.008 ± 0.001 µmol/10^4^ cells, *p* < 0.01) by Tukey’s post hoc test (Figure 3B).

### 3.2. In Vivo Experiment

#### 3.2.1. Animals

A 5 mm mid-diaphyseal defect was created in the femora of 10-week-old male Wistar rats, stabilized with an internal fixator and augmented by a CHS, CBA or MCM scaffold. Defect healing was evaluated at 12 weeks post-surgery. Thirty out of 32 animals survived the surgeries and the observation period. Two animals died intraoperatively, due to unknown reasons (CHS group).

#### 3.2.2. Regenerated Bone Volume (BV)

Defect healing and bone regeneration were evaluated post mortem, using µCT reconstructions. Data showed that the defects that were treated with CBA contained significantly more bone volume (68.2 ± 12.1 mm^3^) than the ones treated with MCM (51.6 ± 12.6 mm^3^) or CHS (44.0 ± 15.4 mm^3^; one-way ANOVA, *p* < 0.0012; Figure 4B). What is more, in the intergroup analysis, by Tukey’s post hoc test, the CBA scaffolds showed a significantly greater bone volume than the MCM (*p* < 0.05) and CHS (*p* < 0.01) scaffolds.

#### 3.2.3. H&E Staining

To assess the grade of defect healing, histological sections that were stained with hematoxylin and eosin were evaluated, according to Huo et al. [13]. Figure 5 shows examples of H&E-stained sections of the experimental groups.

A blinded assessment revealed a mean score of 4.4 ± 1.3 for the implants made of CHS, 4.7 ± 1.0 for MCM, and 5.9 ± 0.8 for CBA. The results showed strong statistical significance (*p* < 0.0001), as determined by one-way ANOVA. Inter-group comparison, using Tukey’s post hoc test, showed significantly better bone healing in the defects treated with CBA scaffolds compared with the CHS and MCM groups (CBA vs. CHS: *p* < 0.001; CBA vs. MCM: *p* < 0.001), as shown in Figure 5C.

#### 3.2.4. Vascularization

Histological enumeration of vessels revealed 147 ± 53 vessels within the defect area of the CHS group, 227 ± 173 for MCM, and 308 ± 233 vessels for CBA (Figure 6B). No relevant overall significance could be determined (one-way ANOVA, *p* = 0.1419). The comparison of the study groups, by means of Tukey’s post hoc test, also did not reveal any statistically significant differences.

#### 3.2.5. Number of Osteoclasts

The MCM group showed an average of 26.9 ± 17.2 osteoclasts per defect area. The CBA groups showed an average of 32.4 ± 19.8, and the CHS group an average of 32.7 ± 19.5 osteoclasts per defect area (Figure 7B).

A comparison of data from the experimental groups yielded no statistically significant differences, in terms of osteoclast numbers (one-way ANOVA; *p* = 0.7474).

#### 3.2.6. Number of Osteoblasts

Within the defined defect area, the MCM group showed an average of 2259 ± 1091, the CBA group 3587 ± 1338, and the CHS group 2115 ± 605 osteoblasts, where the general analysis showed significant differences between the groups (one-way ANOVA; *p* = 0.0077). Bone defects that were treated with CBA scaffolds showed a significantly higher number of osteoblasts (Tukey’s post hoc test; CBA vs. CHS: *p* < 0.05; CBA vs. MCM: *p* < 0.05; Figure 8B).

## 4. Discussion

The aim of our study was to investigate, in vitro and in vivo, the osteoregenerative potential of a scaffold made of the biopolymer chitosan that was produced by the NSN technique. 

When hTERT-MSCs were cultured in vitro in our study, on CHS scaffolds, these cells showed significantly higher osteogenic differentiation—as reflected by the measures of bone-specific alkaline phosphatase (ALP)—when compared to our reference groups CBA and MCM.

This confirms the results of previous studies. For example, Costa-Pinto et al. [17] seeded melt-based chitosan scaffolds with a mouse mesenchymal stem cell line (BMC9), in vitro. They demonstrated that stem cells on chitosan tend to undergo high osteogenic differentiation after an incubation period of 3 weeks, which was associated with increased ALP activity.

Our in vivo study, by examination of rat femora by µCT and histological evaluation, did not show significantly better bone healing in the CHS group when compared to the reference groups. This discrepancy between in vivo and in vitro experiments may be due to the differences in mechanical stability of the scaffolds that were used in this study. The CHS scaffolds exhibit favorable biomimetic properties, in terms of the pore size and porosity of the scaffolds, as described by Rezwan et al. [18]. The authors found that for effective integration of cells and their growth in a scaffold, a pore size of more than 100 µm and a porosity of 90% are required. Human freeze-dried cancellous bone is reported to have pore sizes in the range of 0.1–350 µm. The MCM scaffolds were found to have a mean pore size of about 200 µm [9,19].

However, the mechanical and structural properties of the scaffold material produced with the NSN technique, might have to be adapted to such an extent that the structural integrity is ensured [20]. Ideally, the material properties of the scaffold should match those of the native material, i.e., the bone [21], and be able to withstand physiological loads in vivo. Consequently, the low biomechanical stability of the CHS scaffold might explain the aforementioned differences in these groups. 

One approach to improve the properties of CHS-based scaffolds, regarding mechanical stability and osteoconductivity, is the biochemical modification of the biopolymer. Various methods of modifying the CHS have already been described. For example, it is possible to add other biomaterials (e.g., β-tricalcium sulfate, hydroxyapatite), natural or synthetic polymers (e.g., collagen, silk fibroin, alginate), or bioactive pharmacological molecules (e.g., BMP-2, TGF-β1), to CHS [6].

A similar biochemical modification study, conducted by Zhang et al. [22], showed, in vivo, that bonding nanohydroxyapatite with chitosan improved the mechanical properties of the scaffold. In this study, a critical bone defect was created in the distal femoral condyle of the rabbit, and a nanohydroxyapatite–chitosan composite was injected into the defect. The results were compared to the treatment of a similar bone defect that was treated with pure chitosan. Bone healing was evaluated by µCT and histological examination, 12 weeks after implantation. The experimental group, treated with the nanohydroxyapatite–chitosan composite, showed complete healing of the bone defect, whereas the bone defect in the comparison group, filled with pure chitosan, healed only partially.

The results from in vitro experiments, conducted by Heinemann et al. [23], in which chitosan scaffolds that were produced with the same NSN technique were either coated with collagen type 1 or mineralized with organically modified hydroxyapatite, showed enhanced osteoconductive properties of modified scaffolds compared to scaffolds made of pure, untreated CHS. In both the modified groups, improved cell adhesion, osteogenic differentiation, and proliferation of hBMSCs was observed. The proliferation of osteogenically induced stem cells was the greatest in the group that was treated with organically modified hydroxyapatite, and osteogenic differentiation was highest in the collagen-coated scaffolds. It would be useful to investigate whether these results can be reproduced in vivo.

An advantage of producing chitosan scaffolds using the NSN technique is the possibility of varying lengths and diameters of the fibers used. Additionally, the scaffold can be produced in an individual three-dimensional structure. A change in these parameters is accompanied by a change in both the pore size and porosity [7]. Loh and Choong [20] found that pore size and porosity have a great influence on the mechanical stability of the scaffolds; an increase in porosity is associated with a concomitant decrease in mechanical stability. Varying the parameters of the chitosan fibers represents another approach to produce more mechanically stable scaffolds.

In our study, the fiber-based CHS structures were stabilized with a fibrin gel. In textile technologies and light-weight engineering, fiber–matrix composites are exploited, to adjust compressive and tensile strength and stiffness [24]. Since the fibrin gel that was used has low compressive stability, a matrix system that is adapted to the strength and stiffness properties of bone offers a promising approach to the in vivo investigation of fiber-based scaffold systems. For this purpose, it is necessary to investigate different matrix systems, e.g., based on bone cements, and to study the fiber–matrix interactions in detail. A fundamental design approach, for the realization of such systems combining fiber-based and pasty components, has been laid down by Brünler et al. [25].

Our study has some limitations. In evaluating our in vitro experiment, it is important to note that immortalized hTERT cell lines behave differently under the normal physiological conditions that are found in vivo compared to in vitro, where the cells are exposed to a specific osteogenic medium. Another limitation of this study is the assessment of in vivo bone volume formation in the femora of the CBA group, by µCT. This method cannot differentiate between human cancellous bone and the formed bone of the rat. Computerized subtraction of the known volume of the implanted scaffold is confounded by the fact that the scaffold substance is resorbed by the animal.

## 5. Conclusions

In summary, we conclude that, in vitro, CHS scaffolds promote the osteogenic differentiation of stem cells better than in the control condition, which is an effect that was not observed in vivo.

Because of the biomechanical and biochemical properties of CHS, and the wide range of structures that can be produced with it, using the NSN technique—including processing to thread for weaving and knitting—this method shows promise for the production of scaffolds and other techniques, for bone and soft-tissue reconstruction. Furthermore, to enhance the osteoconductive properties, studies are investigating how processing CHS as a biopolymer or chemical modification would be useful.

## Figures and Tables

**Figure 1 biomedicines-09-01015-f001:**
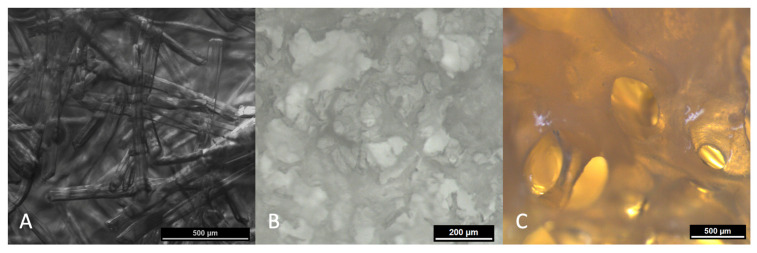
Microstructure of implanted scaffolds: (**A**) chitosan; (**B**) mineralized collagen; (**C**) human cancellous bone. Images were obtained with the encoded stereo microscopes Leica M125 C (Leica Microsystems, Wetzlar, Germany).

**Figure 2 biomedicines-09-01015-f002:**
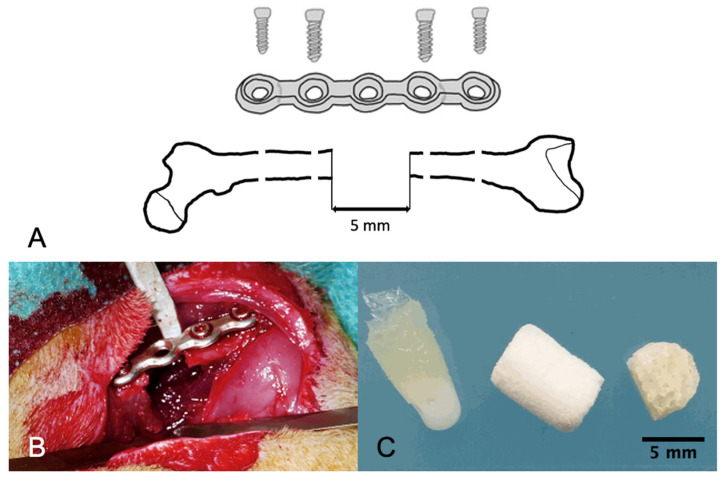
Femoral critical-size bone defect model: (**A**) schematic drawing of the femoral bone defect with internal fixator, (**B**) 5 mm defect at the rat femur stabilized with an internal fixator and (**C**) scaffolds for implantation (from left to right: MCM, CHS, CBA).

**Figure 3 biomedicines-09-01015-f003:**
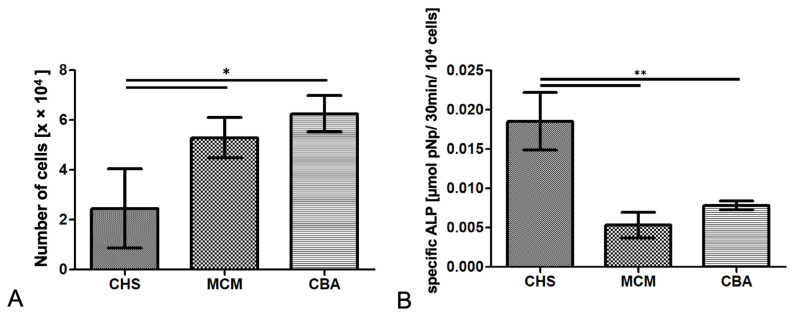
In vitro measurement of (**A**) cell number and (**B**) alkaline phosphatase (ALP) activity after 14 days. Cells of a human mesenchymal stromal cell line (hTERT-MSC) were seeded on the different scaffolds and cultured for 14 days in osteogenic medium to compare cell proliferation and osteogenic differentiation (mean ± SD, *n* = 3; * *p* < 0.05, ** *p* < 0.01; CHS: chitosan, MCM: mineralized collagen, CBA: human cancellous bone allograft).

**Figure 4 biomedicines-09-01015-f004:**
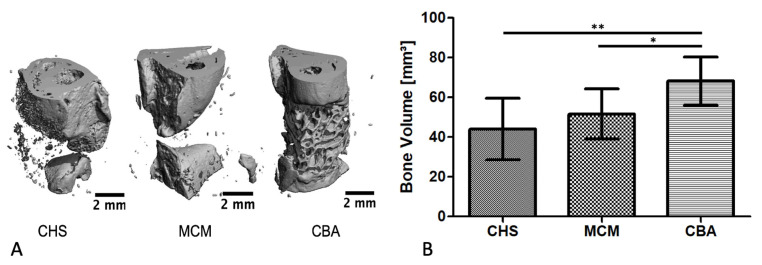
Microcomputed tomography determination of bone volume at the defect side after a 12-week follow-up. (**A**) Representative 3D reconstructions of the defect area in the different groups and (**B**) quantitative analysis of the regenerated bone volume (mean ± SD; CHS: *n* = 9, MCM: *n* = 10, CBA: *n* = 11; * *p* < 0.05, ** *p* < 0.01).

**Figure 5 biomedicines-09-01015-f005:**
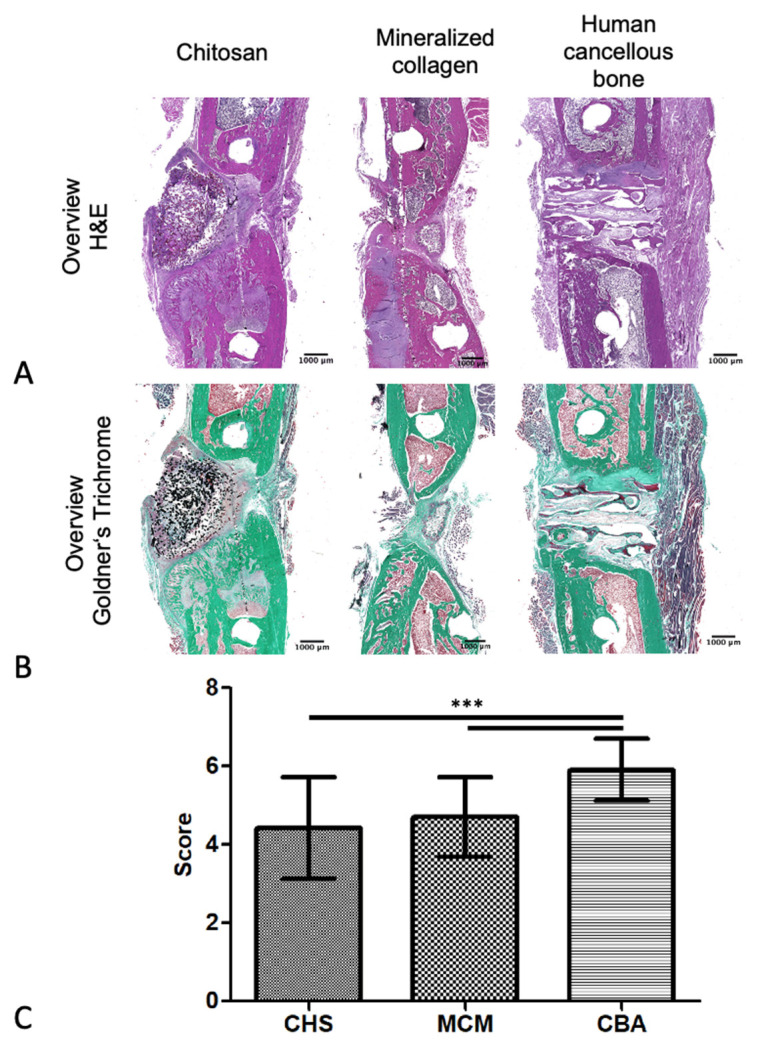
Histomorphological scoring of the hematoxylin and eosin (H&E)-stained defect areas according to Huo et al. (**A**) Representative sections for each group stained with H&E; (**B**) representative sections for each group stained with Goldner’s Trichrome (green color demonstrates mineralized bone tissue; pink/red stained segments indicate non mineralized fibrous tissue, bone marrow or muscle); (**C**) grade of defect healing (mean ± SD; CHS: *n* = 9, MCM: *n* = 10, CBA: *n* = 11; *** *p* < 0.001).

**Figure 6 biomedicines-09-01015-f006:**
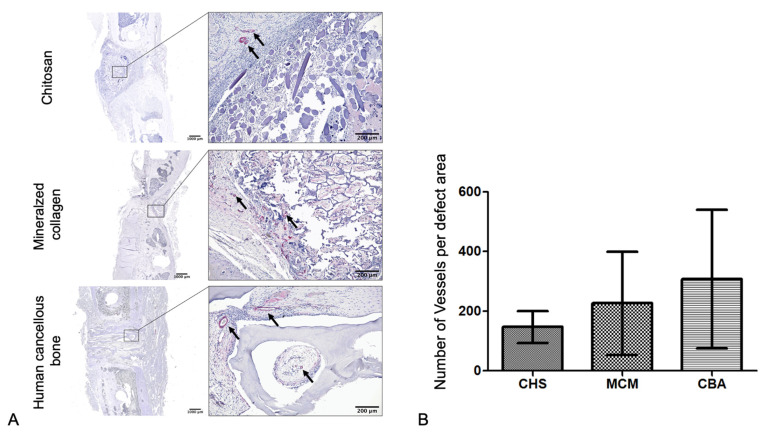
Vascularization of the defect area after 6 weeks as demonstrated by α-smooth muscle actin immunostaining. (**A**) Representative histological sections (black arrows depict vessels) of the three groups and (**B**) quantitative analysis by counting the vessels present in the defect area (mean ± SD, CHS: *n* = 9, MCM: *n* = 10, CBA: *n* = 11).

**Figure 7 biomedicines-09-01015-f007:**
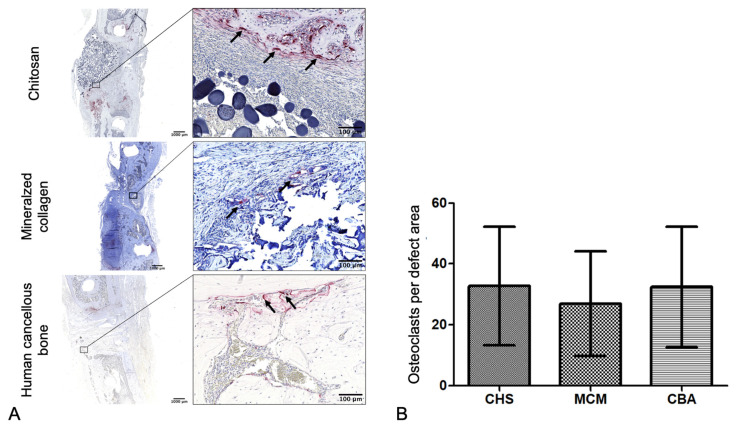
Immunohistochemical staining of osteoclasts. (**A**) Representative histological sections of tartrate-resistant acid phosphatase-positive osteoclasts (arrows) of each group and (**B**) quantitative analysis by counting the osteoclasts present in the defect area (mean ± SD, CHS: *n* = 9, MCM: *n* = 10, CBA: *n* = 11).

**Figure 8 biomedicines-09-01015-f008:**
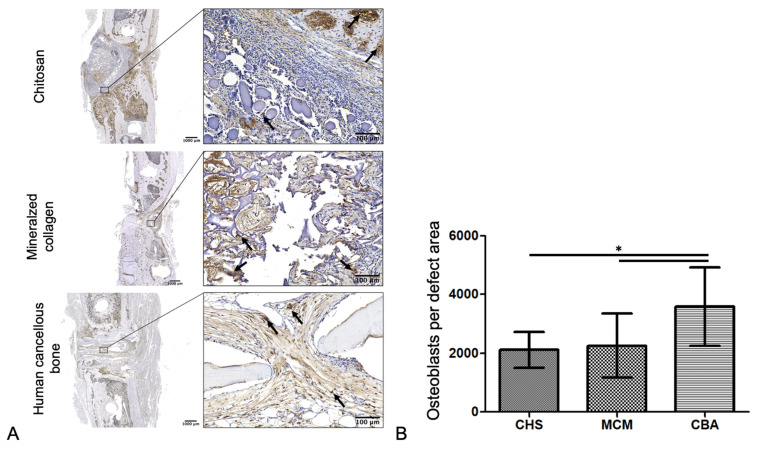
Immunohistochemical staining of osteoblasts. (**A**) Representative histological sections of alkaline phosphatase-positive osteoblasts (arrows) of each group and (**B**) quantitative analysis by counting the osteoblasts present in the defect area (mean ± SD, CHS: *n* = 9, MCM: *n* = 10, CBA: *n* = 11; * *p* < 0.05).

**Table 1 biomedicines-09-01015-t001:** Score according to Huo et al., for evaluation the grade of defect healing.

Score	Associated Findings at Fracture Site
1	fibrous tissue
2	predominantly fibrous tissue with small amounts of cartilage
3	equal parts of fibrous and cartilaginous tissue
4	predominantly cartilaginous tissue with small amounts of fibrous tissue
5	cartilage
6	predominantly cartilage with small amounts of immature bone
7	equal parts of cartilage and immature bone
8	predominantly immature bone with small amounts of cartilage
9	union of fracture fragments with immature bone
10	union of fracture fragments with mature bone

## Data Availability

The primary data supporting the reported results can be provided upon request by the corresponding author.

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
