# Peer review of "Treatment of Critical-Size Femoral Bone Defects with Chitosan Scaffolds Produced by a Novel Process from Textile Engineering"

_biomedicines, 2021, doi:10.3390/biomedicines9081015_

Round 1
Reviewer 1 Report
The text was well written and interest. It is better to show the higher magnification in immunohistochemical studies. in addition, regarding vascularization, the data of VEGF and CD31 contributes this study to provide stronger impact.
The ideal bone graft material is demanded in various fields. Consequently, its development is crucial. In this respect, this study has a meaningful.
Pore sizes should be matched in each scaffold for comparing different materials. The ideal size for cell infiltration into scaffold has been already reported as 150-500 mm.
Table 1 shows the score of bone formation, but it should be stated whether there is objectivity in this evaluation method, and if so, how to prove it.
Evaluation of new bone volume by micro-CT is important, but it is important to describe how the scaffold and new bone were distinguished.
CBA has the highest bone formation in the histology, and there is little evidence that CHS is useful.
Reviewer 2 Report
Dear Authors, below are my comments about the submitted manuscript.
- The title of the manuscript well conveys with the major concern of the study.
- The abstract is well structured and properly summarize the topic addressed.
- The references are up to date.
- The Introduction section must be improved in my opinion. It is too short, and it can better introduce the development of the manuscript if modified. I think that periods in lines 311-324 page 11 can be moved from Discussion to Introduction section. The aim of the study is properly expressed.
- Pag. 3 lines 95-96: please indicate the autoclaving setting and procedure used.
- Figure 1: I cannot find an in-text citation of the figure. Moreover, figure caption needs to be more exhaustive: you must indicate where the images were taken from (microscope) and the magnification used.
- Why did you not perform the sample size calculation? Did you base on previously published study? Please clarify.
- I would like to know why you did not conduct the study biochemically modifying the structure of the CHS as reported by, for example, Zhang et al (21) and Heinemann et al. (22) that you cited. I mean, why did you try to test in vivo behaviour of pure CHS if there are previous results demonstrating that modifying its structure adding different molecules improve its properties, most of all mechanical? It could be an unnecessary scarification of animals under test. Please clarify. Results in this present condition are pretty discouraging, even if significant.
Round 2
Reviewer 2 Report
No any other comments